# Assessing Inhaler Techniques of Asthma Patients Using Aerosol Inhalation Monitors (AIM): A Cross-Sectional Study

**DOI:** 10.3390/healthcare11081125

**Published:** 2023-04-13

**Authors:** Mansour M. Alotaibi, Louise Hughes, William R. Ford

**Affiliations:** 1Pharmacy Practice Department, College of Clinical Pharmacy, King Faisal University, Al-Ahsa 31982, Saudi Arabia; mmqalotaibi@kfu.edu.sa; 2School of Pharmacy and Pharmaceutical Sciences, Cardiff University, Cardiff CF10 3NB, UK

**Keywords:** asthma, inhaler technique, pMDI, MDI, DPI, spacer, management, inhaler device, aerosol inhalation monitor, Vitalograph

## Abstract

A high percentage of asthma patients have symptoms that are not well controlled, despite effective drugs being available. One potential reason for this may be that poor inhaler technique limits the dose delivered to the lungs, thereby reducing the therapeutic efficacy. The aim of this study was to assess the prevalence of poor inhaler technique in an asthma patient population and to probe the impact of various demographic parameters on technique quality. This study was conducted at community pharmacies across Wales, UK. Patients diagnosed with asthma and 12 years or older were invited to participate. An aerosol inhalation monitor (AIM, Vitalograph^®^) was used to measure the quality of patient inhaler technique. A total of 295 AIM assessments were carried out. There were significant differences in the quality of inhaler technique across the different inhaler types (*p* < 0.001, Chi squared). The best technique was associated with dry-powder inhalers (DPI devices, 58% of 72 having good technique), compared with pressurized metered-dose inhalers (pMDI) or pMDIs with a spacer device (18% of 174 and 47% of 49 AIM assessments, respectively). There were some significant associations between gender, age, and quality of inhaler technique, as determined with adjusted odds ratios. It seems that the majority of asthmatic patients were not using their inhalers appropriately. We recommend that healthcare professionals place more emphasis on assessing and correcting inhaler technique, as poor inhaler technique might be responsible for the observed lack of symptom control in the asthma patient population.

## 1. Introduction

The National Review of Asthma Deaths (NRAD) UK (2014) reported that 46% of deaths from asthma were preventable [1]. Allied to this report, studies have reported a high percentage of asthma patients with symptoms that are not well controlled [2,3,4,5]. There are numerous factors that contribute to the lack of symptom control and preventable mortality in asthma; these involve both healthcare professionals and patients. In terms of patient factors, there are theoretically two major patient factors that might influence control of asthma symptoms. The first is inhaler technique, and the second is adherence to prescribed medication. Studies have demonstrated the importance of inhaler technique and patient adherence in improving asthma outcomes [2,6]. Therefore, it is probable that poor inhaler technique and adherence contribute to lack of symptom control and preventable mortality in the asthma patient population. In this study, we focused on patient inhaler technique.

Previous studies reported that somewhere between 28% and 68% of patients who use pressurized metered-dose inhalers (pMDIs) or dry-powder inhalers (DPIs) do so incorrectly [7]. Poor inhaler technique means that patients may receive subtherapeutic doses, adversely affecting their quality of life and increasing the risk of morbidity and mortality [2,7]. Theoretically, this is a particular problem for inhaled corticosteroids, due to the lack of physiological feedback informing patients that a sufficient dose has been delivered. For reliever inhalers, an effective dose can be sensed through the rapid relief of asthma symptoms.

Although inhaler technique has previously been studied in patient populations, there was a large degree of variability in the extent of improper technique reported [2,3,7]. It is possible that this was due to the subjective nature of the assessments used to determine the quality of inhaler technique. Therefore, we used a more quantitative approach, based on the measurement of inspiratory flow, to assess the different inhaler techniques used for pressurized metered-dose and dry-powder inhalers, and using an aerosol inhalation monitor (AIM, Vitalograph^®^, Maids Moreton, UK). This is an interactive tool that enables healthcare providers to objectively assess patient inhaler technique. An AIM can classify patient inhaler techniques across different inhaler device types into three categories of good, suboptimal, and poor, based on four essential components of good inhaler technique: canister activation, inspiratory flow rate, breath hold time, and inhalation time [8]. The quality of technique is assessed as the predicted pulmonary drug deposition, based on aspects of good inhaler technique. This is displayed as a pictorial representation to the patient of where the drug is deposited in their lungs. The categories of good, suboptimal, and poor are defined as drug deposition in the small airways, large airways, or oropharyngeal area, respectively [9]. Furthermore, associations between some patient demographics and quality of inhaler technique have previously been reported. The frequency of poor pMDI technique increases as people get older [2]. It was also observed that poor coordination was more common in females than males [2].

### Aims and Objectives

The aim of this study was to assess the prevalence of poor inhaler technique in an asthma patient population. The research had two objectives: first, to investigate how asthma patients handled inhalers across three different inhaler devices; and second, to assess the impact of various demographic parameters and smoking status on technique quality, to try to identify any factors that might be associated with poor inhaler technique.

## 2. Materials and Methods

This was a cross-sectional study, with eight community pharmacies in South Wales selected to recruit patients. A list of postcodes relating to areas with different levels in the index of multiple deprivation were identified. The pharmacies were selected to cover a range of socioeconomic levels, based on the areas they served. Pharmacies were located in Penarth, Newport (two pharmacies), Pontypool, Trevethin, Caerphilly, Cathays, and Blackwood. Any asthma patient older than 12 years could participate in the study. Non-probability convenience sampling was used to recruit patients [10]. Patients were invited to participate in the study in two ways. First, they were invited verbally at the time of dispensing of asthma medications in the selected pharmacies. Second, patients were identified from pharmacy databases and then contacted by phone. They were given a participant information sheet and written informed consent was obtained prior to taking part in the study. Approval was obtained from the Research Ethics Committee of the Cardiff School of Pharmacy and Pharmaceutical Sciences (1617-01).

Data collection took place in the period from October 2016 to July 2018. Patients were interviewed in a consultation room. Once consent was obtained, a standardised form was used for data collection. The form had three main sections: participant demographic information, clinical information (e.g., information about asthma and medications), and the result of the technique assessment carried out using an aerosol inhalation monitor (AIM, VItalograph^®^, Maids Moreton, UK). Once the first two sections were completed by the assessors, patients were asked to demonstrate their inhaler technique using an AIM based on the patient’s usual inhaler device type(s). Each participant was given a unique identification number based on the location of the community pharmacy. Identifying information was kept separately from the study data in secured databases.

### Data Analysis

Data analysis was carried out using IBM SPSS^®^ Statistics version 25 software. Frequencies and descriptive statistics were calculated to obtain a general view of the patient demographic variables, smoking status, and AIM assessments across inhaler devices. All variables were treated as nominal variables, except for the dependent variable (i.e., AIM assessments), which was treated as an ordinal. Asthma patients were divided into three groups, based on the different types of inhaler used (1 = pMDI, 2 = pMDI with a spacer, 3 = DPI). Some patients used more than one type of inhaler device, so the analysis was performed based on the total number of AIM assessments that were carried out across inhaler devices.

To assess the association between demographic variables and the quality of inhaler technique with the AIM device, the data were analysed using multinomial logistic regression (MLR). Three MLR models were utilized to obtain the odds ratios and associated *p*-values. The inhaler devices were stratified into three groups (pMDI, pMDI with a spacer, DPI). Then, a univariate MLR was performed for each device. The assumptions of the MLR analysis were checked and there were no violations. Adjusted odds ratios with a 95% confidence interval (95% CI) for independent variables were obtained to assess the relationship between independent and dependent variables. For all statistical tests, a *p*-value of 0.05 was considered to be statistically significant, and all statistical tests were two-tailed.

## 3. Results

### 3.1. Participant Demographics

Two-hundred and twelve patients participated in the study between October 2016 and July 2018. Their ages ranged from 16 to 91 years (*m =* 57.1, SD ± 19.1, Mdn = 61 years). Of those who participated in the study, 113 (53%) participants were females (*m* = 53.8 years, SD ± 19.8) and 99 (47%) participants were males (*m =* 60.8 years, SD ± 17.2). The majority of participants were younger than 65 years (*n* = 124, 59%). Most of the participants were non-smokers (*n* = 180, 85%), with 12 smokers out of 99 and 20 smokers of out of 113 for males and females, respectively. The total number of inhaler technique assessments was 295, as 75 participants (35%) were using more than one type of inhaler device. In total, 174, 49, and 72 AIM assessments were carried out for pMDI, pMDI with a spacer, and DPI, respectively (Table 1).

### 3.2. Evaluation of Inhaler Technique

There were significant differences in the quality of inhaler technique across the three inhaler types (*p* < 0.05, Chi-squared test). The best inhaler technique was associated with DPI simulators, for which 42 AIM assessments (58%) out of 72 were good, while poor inhaler technique was most often seen with the pMDI simulators (Table 1). The results of the post hoc test showed that four groups were statistically different. Good inhaler technique was more strongly associated with DPI simulators, which also had fewer poor results (*p* < 0.0057). Conversely, pMDI simulators were more strongly associated with poor inhaler technique and had fewer good results (*p* < 0.0057). No statistical differences were associated with using pMDIs with spacers nor with suboptimal outcomes.

### 3.3. The Impact of Demographic Variables and Smoking Status on Inhaler Technique

There were no statistically significant associations between gender and the quality of inhaler technique across the different devices. Age was significantly associated with the quality of technique for patients using a pMDI with a spacer (*p* < 0.02, Fisher’s exact test). For the other devices (DPI and pMDI), no relationship was found between age and quality of inhaler technique (Table 1, Table 2, Table 3 and Table 4). The probability of having good, suboptimal, or poor inhaler technique significantly varied based on the independent variables, as determined with the adjusted odds ratios (AOR). For pMDI simulators, men were 2.6 times more likely to have a good technique compared to women (AOR 2.6, 95% CI: 1.121–6.111; *p* < 0.05: see Table 5). For pMDI with a spacer simulators, participants younger than 65 years were less likely to have suboptimal results (AOR 0.024, 95% CI: 0.002–0.331; *p* < 0.05: see Table 5). Men were also less likely to have a suboptimal technique with pMDI with a spacer devices in comparison to women (AOR 0.097, 95% CI: 0.011–0.870; *p* < 0.05: see Table 5). There was no potential association between smoking status and quality of inhaler technique. For the DPI simulators, no relationship was found between performance on the AIM device and the independent variables (Table 5).

## 4. Discussion

Studies have shown that the majority of asthma patients do not use their inhaler devices properly [2,3,7,11]. In these studies, there was a wide variation in the percentage of patients who had improper inhaler technique. In the previous studies, inhaler technique was generally assessed through observation rather than empirical measurement [2,3,7,11]. As it can be hard to estimate inspiratory flow rates and timing of activation by simple observation, they may have overestimated the quality of the inhaler technique. Therefore, there was a need to assess patients’ inhaler techniques via an objective quantitative assessment tool. We used an AIM device to assess inhaler technique in asthma patients. To the best of our knowledge, this study is the first to assess patients’ inhaler technique and the impacts of demographic variables across all types of inhaler device and using an objective quantitative assessment tool.

After assessing patients’ inhaler technique, it seemed that the majority of asthma patients were not using their inhalers appropriately. Sixty-seven percent of participants in the cohort had poor or suboptimal inhaler technique, as assessed using an AIM device across the three inhaler devices. This is of considerable concern, since a good inhaler technique is very important for delivering a therapeutic dose to the small airways [2,7]. Participants were more likely to use the DPI device appropriately in comparison to the pMDI with a spacer and pMDI devices. This finding was also reported by other authors [3]. Although the best inhaler technique was associated with DPI devices, there are some circumstances in which using a DPI is not advisable, such as in patients who are unable to produce sufficient inspiratory flow, such as children or some elderly people [12]. Furthermore, some adult asthma patients may face difficulty in using DPI devices during an asthma attack, due to the requirement for a fast inhalation compared to the other two device types [7]. Therefore, using DPIs as the only inhaler device for managing all groups of asthma patients is not recommended. It is noteworthy that the resistance to flow is crucial with DPIs, whereas technique is more important for pMDIs. By taking that into consideration, it is more likely that a dispensed inhaler would fit with a patients’ ability to generate adequate inspiratory flow. This could be facilitated using training devices (e.g., AIM and In-Check Dial devices), to assure that patients could use their inhalers properly.

This study showed that the probability of using a pMDI properly was higher in men in comparison to women. Chorão et al., [13] in their cross-sectional observational study, also found that females were more likely than males to incorrectly use a specific type of pMDI inhaler device [13]. Even though the relationship between gender and pMDI usage in the present study was not significantly different (*p* = 0.079), this may have been due to the sample size being too small. This was because participants were classified into different categories (i.e., good, suboptimal, poor, men and women) and this led to having relatively few participants in each group, making extrapolation of the statistical analysis quite difficult. However, when all independent variables were included in the multivariate MLR model for the pMDI device, gender was found to significantly affect the quality of inhaler technique. Men were 2.6-times more likely to have a good inhaler technique with pMDI devices in comparison to women (*p* < 0.05).

There are several electronic devices that could be used to assess patients’ inhaler technique [8]; however, the AIM device is the only one that can be used to assess patients’ inhaler technique across the three different inhalers. Carpenter et al. (2017) conducted a recent study aimed at reviewing multiple electronic devices available on the market that assess patients’ inhaler technique and provide feedback [8]. The results they provided indicated that the AIM device was the only device that could be used across the three types of inhaler (i.e., pMDI, pMDI with spacers, and DPI). The latest version of the In-check DIAL (i.e., G16 In-check DIAL) could be used to evaluate the handling of 16 inhalers on the market via assessing patients’ inspiratory effort [14]. This device is built to recreate the resistance of the 16 inhaler devices (i.e., gentle inspiratory effort for pMDIs and fast for DPIs) [14]. Although this device might be more accurate in assessing the inspiratory effort needed, as the intrinsic resistance of an inhaler device changes in comparison to the AIM, it does not assess the other essential factors required for a proper inhaler technique (i.e., correct canister activation, inhalation times, and breath-hold times) [8,14].

Health care providers should pay attention to patients’ inhaler technique. However, even though asthma management guidelines stress the importance of checking inhaler technique, the majority of asthma patients in our study were not using their inhaler devices properly. Health care providers should pay more attention to pMDI users in particular. This is because pMDI devices are associated with the worst inhaler device technique, based on the current findings and supported by previous research [3,15]. They should also take into consideration the variation in the pMDI technique among people with different genders. Furthermore, prescribing the type of inhaler that matches patients’ ability to produce an adequate inspiratory flow rate might improve asthma control. This is because having more than one type of inhaler device would increase the probability of improper inhaler device usage [11]. On the other hand, although the probability of having suboptimal results with the pMDI with a spacer device was statistically significant with respect to age and gender, this is probably not clinically significant. This is because demonstrating a proper inhaler technique (i.e., not suboptimal) is essential in asthma control [16]. Furthermore, several studies have shown that inappropriate inhaler technique was associated with poor asthma control [2,7].

### Study Strengths, and Limitations

Variability is likely to be introduced in the assessment of patients’ inhaler technique using a checklist, due to the subjective nature of the non-empirical measurements made by assessors. It is difficult to accurately assess certain essential features of inhaler technique, such as the inspiratory flow rate and excessive deposition on the tongue, solely by observation. Our study assessed patients’ inhaler technique using an objective quantitative device, which has been approved for assessing inhaler technique [8]. Using an objective tool yields more accurate and consistent results. Although, some features of technique still required non-empirical observation, such as whether a MDI was shaken prior to use, these aspects of technique are easier for an investigator to characterise, being binary in nature.

There is an important limitation in using an AIM to assess the quality of DPI technique, which is that it has only one resistance level for all DPI devices [17]. The DPI devices currently on the market have widely varying airflow resistances and require different inspiratory flow rates [7,18]. They are classified into three categories based on their intrinsic resistance: low, moderate, and high resistance DPIs [18]. Intrinsic resistance to flow is directly associated with the degree of inspiratory effort required to generate sufficient inspiratory flow for dose delivery [18,19,20]. Therefore, the fact that the AIM-simulated DPI does not replicate the different resistances of the specific inhaler used by a patient is a limitation of our study. However, the resistance to flow of the AIM-simulated DPI is classified as moderate [17]. This may lead to underestimation of the quality of technique for low-resistance DPI devices and overestimation for high-resistance DPI devices. That being said, there is a minimum required inspiratory flow rate, which is called the “acceptable inhalation rate”, for each type of DPI device [17]. In a study that aimed to determine the inspiratory flow rate limits of inhalers, most of the DPI devices on the market were found to have an acceptable inspiratory flow rate, ranging from 30–35 L/min “moderate resistance” [17]. As an increase in intrinsic resistance would necessitate a fast-inspiratory flow rate [14], DPIs with a moderate intrinsic resistance may not require such a fast flow rate as DPIs with high intrinsic resistance. Therefore, knowing that the AIM DPI simulator has only a moderate level of resistance (i.e., the same level associated with an acceptable inspiratory flow rate) might indicate that the AIM could be used to assess the quality of inhaler technique for a wide range of DPI devices. Furthermore, the AIM device does not capture all elements of inhaler technique. For example, good MDI technique requires that the device is shaken prior to dose delivery. This is not incorporated into the AIM assessment of quality. Therefore, it is likely that our assessment of MDI technique is an overestimate of the quality of technique for the asthma patient population of Wales.

Convenience sampling was used to recruit participants from several different community pharmacies in Wales, to try and ensure good representation of the patient population. The patient cohort was recruited from different geographic areas (areas with different socioeconomic status) and included female and male patients of different ages. Therefore, the sample was more likely to be representative of the whole population. However, the nature of convenience sampling likely means that the recruited cohort of patients was more adherent than the general population, as they were willing to engage in the study. The likelihood is that our study overestimated the quality of inhaler technique in this asthma patient population.

The dependent variable (i.e., the quality of inhaler technique) was treated as a nominal variable instead of an ordinal one in the MLR model. All statistical tests that were performed (e.g., Chi-square and MLR) could be conducted on both variables (i.e., nominal or ordinal). It has been claimed that MLR could be run for an ordinal variable without violating any assumptions, as long as the researcher is able to answer the research question [21,22]. The appropriateness of using MLR analysis with an ordinal variable might be impacted as the number of independent variables increases or as the categories of dependent variables increase [23]. Given that there were only a few independent variables and categories, an MLR analysis was conducted. Furthermore, it would be preferable to use multilevel multinomial logistic regression and consider the patient as the unit of analysis, especially when it comes to examining the impact of the number of inhaler devices used per patient on the quality of inhaler technique. However, this analysis was not performed, as it would have meant many subgroups contained a very low number of patients, due to the heavily skewed distribution towards poor inhaler technique. Thus, it was decided to use MLR analysis and consider AIM assessment as the unit of analysis, to overcome the issue of having very few participants in several subgroups.

In this study, we chose to focus on inhaler technique. However, there is an additional patient-centric factor that is likely to be important in determining control of asthma symptoms. Patient adherence, particularly with preventer medication, is likely to have an additional important impact on the control of asthma symptoms in patients. In this study, we did not study whether patients adhered to their prescribed preventer medication or not, but plan to address this in future studies.

## 5. Conclusions

This study set out to determine the inhaler technique amongst asthma patients and discovered that the inhaler technique in the asthma patient population is poor. In particular, inhaler technique associated with pMDIs was particularly poor, with women tending to have worse technique than men. A deficit in inhaler technique is likely to impact on the optimal drug delivery to the lungs and may contribute to therapeutic failure, with a resultant lack of symptom control. Based on these findings, healthcare professionals urgently need to place more emphasis on assessing and improving inhaler technique in asthma patient populations.

## Figures and Tables

**Table 1 healthcare-11-01125-t001:** Inhaler device and AIM result cross tabulation (295 AIM assessments). *p*-value is <0.05.

Inhaler Device	AIM Result			Total
	Poor	Suboptimal	Good	
pMDI	102 (58.6%)	40 (23%)	32 (18.4%)	174
pMDI + spacer	12 (24.5%)	14 (28.5)	23 (47%)	49
DPI	7 (9.7%)	23 (32%)	42 (58.3%)	72
Total	121	77	97	295

**Table 2 healthcare-11-01125-t002:** Relationship between AIM pMDI simulators and demographic variables (*n* = 174).

N = 174 AIM Assessments					
Variable		pMDI Inhaler Device			*p*-Value
		Poor (Freq/%)	Suboptimal (Freq/%)	Good (Freq/%)	
**Age**	Less than 65 years	65 (37.4%)	20 (11.5%)	22 (12.6%)	0.206
	65 years and older	37 (21.3%)	20 (11.5%)	10 (5.7%)	
**Gender**	Male	45 (25.9%)	17 (9.8%)	21 (12.1%)	0.079
	Female	57 (32.8)	23 (13.2%)	11 (6.3%)	
**Smoking**	Non-smoker	90 (51.7%)	32 (18.4%)	25 (14.4%)	0.260
	Smoker	12 (6.9%)	8 (4.6%)	7 (4%)	

**Table 3 healthcare-11-01125-t003:** Relationship between AIM pMDI with a spacer simulators and demographic variables (*n* = 49).

N = 49 AIM Assessments					
Variable		pMDI with a Spacer Inhaler Device			*p*-Value
		Poor (Freq/%)	Suboptimal (Freq/%)	Good (Freq/%)	
**Age**	Less than 65 years	10 (20.4%)	4 (8.2%)	13 (26.5%)	0.019
	65 years and older	2 (4.1%)	10 (20.4%)	10 (20.4%)	
**Gender**	Male	6 (12.2%)	4 (8.2%)	14 (28.6%)	0.199
	Female	6 (12.2%)	10 (20.4%)	9 (18.4%)	
**Smoking**	Non-smoker	8 (16.3%)	12 (24.5%)	20 (40.8%)	0.313
	Smoker	4 (8.2%)	2 (4.1%)	3 (6.1%)	

**Table 4 healthcare-11-01125-t004:** Relationship between AIM DPI simulators and demographic variables (*n* = 72).

N = 72 AIM Assessments					
Variable		DPI Inhaler Device			*p*-Value
		Poor (Freq/%)	Suboptimal (Freq/%)	Good (Freq/%)	
**Age**	Less than 65 years	3 (4.2%)	9 (12.5%)	18 (25%)	0.938
	65 years and older	4 (5.6%)	14 (19.4%)	24 (33.3%)	
**Gender**	Male	3 (4.2%)	10 (13.9%)	21 (29.2%)	0.939
	Female	4 (5.6%)	13 (18.1%)	21 (29.2%)	
**Smoking**	Non-smoker	6 (8.3%)	22 (30.6%)	35 (48.6%)	0.354
	Smoker	1 (1.4%)	1 (1.4%)	7 (9.7%)	

**Table 5 healthcare-11-01125-t005:** Adjusted and unadjusted odds ratios table and corresponding confidence intervals.

AIM Assessment	Variable	Unadjusted OR [95%CI]	Adjusted OR [95%CI]
**pMDI (suboptimal)**	Sex (ref = women) Men	0.93 [0.44–1.96]	0.84 [0.39–1.80]
	Age (ref = 65 years and older) Less than 65 years	0.56 [0.27–1.19]	0.48 [0.22–1.06]
	Smoking Status (ref = smoker) Non-smoker	0.53 [0.20–1.42]	0.43 [0.15–1.20]
**pMDI (good)**	Sex (ref = women) Men	2.41 [1.05–5.53]	2.61 [1.12–6.11] *
	Age (ref = 65 years and older) Less than 65 years	1.25 [0.53–2.92]	1.33 [0.55–3.24]
	Smoking Status (ref = smoker) Non-smoker	0.47 [0.17–1.33]	0.46 [0.16–1.34]
**pMDI with spacers (suboptimal)**	Sex (ref = women) Men	0.40 [0.07–2.02]	0.09 [0.01–0.87] *
	Age (ref = 65 years and older) Less than 65 years	0.08 [0.01–0.54]	0.02 [0.002–0.331] *
	Smoking Status (ref = smoker) Non-smoker	3.00 [0.44–20.43]	0.68 [0.06–7.71]
**pMDI with spacers (good)**	Sex (ref = women) Men	1.55 [0.38–6.35]	0.95 [0.19–4.58]
	Age (ref = 65 years and older) Less than 65 years	0.26 [0.04–1.46]	0.32 [0.04–2.28]
	Smoking Status (ref = smoker) Non-smoker	3.33 [0.60–18.37]	2.22 [0.36–13.55]
**DPI (suboptimal)**	Sex (ref = women) Men	1.02 [0.18–5.66]	1.01 [0.17–5.91]
	Age (ref = 65 years and older) Less than 65 years	0.85 [0.15–4.76]	1.02 [0.16–6.50]
	Smoking Status (ref = smoker) Non-smoker	3.66 [0.19–67.65]	3.71 [0.18–76.00]
**DPI (good)**	Sex (ref = women) Men	1.33 [0.26–6.70]	1.35 [0.25–7.14]
	Age (ref = 65 years and older) Less than 65 years	1.00 [0.19–5.03]	1.03 [0.17–5.98]
	Smoking Status (ref = smoker) Non-smoker	0.83 [0.08–8.04]	0.83 [0.07–9.08]

* *p*-value < 0.05.

## Data Availability

Data are available upon request from the corresponding author. The data from this study are not publicly available, due to the potential that participants might be identified.

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
