# Peer review of "Assessing Inhaler Techniques of Asthma Patients Using Aerosol Inhalation Monitors (AIM): A Cross-Sectional Study"

_healthcare, 2023, doi:10.3390/healthcare11081125_

Round 1
Reviewer 1 Report
The study is very comprehensive. I suggest the following changes be made.
Introduction: Line:49-50- Please describe shortly whether those 3 studies found any association or not.
In addition to age, gender, and smoking, were any additional variables taken into consideration which may affect the outcome of the study?
Were the participants using the same inhalers in their daily life as were used in this study?
Demographic information needs to be in more detail, what was the age median etc. etc. for example- apart from Describing someone’s age as less than 65 years old, reporting the actual age will be worthwhile.
Result- After adjusting the smoking effect, the results show that smoking does not affect the outcome. However, men were found to use inhalers properly as compared to women, therefore, description of male female age and smoking status would be useful.
Discussion- Please describe limitations of the methods used in other studies.
References: Reference number one should be in more detail. Include link if possible.
Reviewer 2 Report
Dear Editor
The study is well-designed and the authors have identified and acknowledged limitations of the study, which is important for the interpretation of the results. The findings may have important implications for the management of patients with asthma, particularly in terms of improving inhaler technique and hence the efficacy of inhaler therapy.
Abstract
The abstract provides a clear and concise overview of the study's aims, methods, and results. The language used is technical but still understandable to a reader with a basic understanding of the subject matter. The abstract effectively highlights the importance of proper inhaler technique in managing asthma symptoms and emphasizes the need for healthcare professionals to assess and correct inhaler technique to improve patient outcomes.
One potential limitation of the study, however, is that it was conducted only in community pharmacies in Wales, which may not be representative of the broader asthma patient population. Additionally, the abstract does not provide information on the specific demographic factors that were found to be associated with inhaler technique quality or the magnitude of these associations.
Introduction:
Overall, this is a well-written and informative piece of writing. The author clearly presents the issue of poor inhaler technique as a contributing factor to the lack of symptom control and preventable mortality in asthma patients. They provide relevant background information on the high percentage of asthma patients with uncontrolled symptoms and the preventable nature of many asthma deaths. Additionally, they highlight the importance of inhaler technique and patient adherence in improving asthma outcomes and reducing preventable mortality.
The author uses citations to support their claims, providing evidence from previous studies to demonstrate the prevalence of poor inhaler technique among patients and the impact this can have on treatment efficacy and patient outcomes. The use of the Aerosol Inhalation Monitor (AIM) to objectively assess patient inhaler technique is a useful addition to the study, providing a more quantitative approach to measuring technique quality.
The aims and objectives of the study are clearly stated, providing a roadmap for the research to follow. The two objectives, investigating how patients handle inhalers across different devices and assessing the impact of various demographic parameters and smoking status on technique quality, are relevant and will likely provide valuable insights into the factors associated with poor inhaler technique.
Overall, this passage is clear, concise, and well-supported, providing valuable information on the importance of inhaler technique in managing asthma and reducing preventable mortality.
Methods
This passage is well-written and provides clear information on the statistical analysis used in the study. The use of Multinomial Logistic Regression (MLR) to assess the association between demographic variables and the quality of inhaler technique on the AIM device is appropriate and is a common approach used in medical research.
The stratification of inhaler devices into three groups (pMDI, pMDI with a spacer, DPI) is a logical way to account for potential differences in inhaler technique across different devices. The use of a univariate MLR for each device is appropriate and allows for the identification of device-specific factors that may be associated with poor inhaler technique.
The author provides information on the assumptions of MLR analysis being checked and no violations being present, indicating that the analysis is sound. The use of adjusted odds ratios with a 95% confidence interval (95% CI) to assess the relationship between independent and dependent variables is appropriate and provides a measure of the strength of the association.
The use of a p-value of 0.05 to determine statistical significance is a common convention in medical research and is appropriate here. The mention of all statistical tests being two-tailed is a useful addition, indicating that the analysis is considering both the possibility of positive and negative associations between variables.
Overall, this passage is clear and provides important information on the statistical analysis used in the study. The author has provided relevant details on the methodology and the statistical approach used, which will be useful for readers in understanding the results and conclusions of the study.
Discussion:
Based on the information provided, the study appears to be a well-conducted investigation of inhaler technique in patients with asthma. The authors have used an objective tool to assess inhaler technique, which may be more reliable and accurate than subjective assessments. However, there are limitations in using the AIM to assess DPI technique, as it does not replicate the different resistances of the specific inhaler used by a patient. This may lead to underestimation of the quality of technique for low resistance DPI devices and overestimation for high resistance DPI devices. The authors acknowledge this limitation and suggest that the AIM could still be used to assess the quality of inhaler technique for a wide range of DPI devices.
The authors have also used convenience sampling to recruit participants from different geographic areas and with different socioeconomic statuses, which may increase the generalizability of the findings. However, the nature of convenience sampling means that the recruited cohort of patients may be more adherent than the general population, which may overestimate the quality of inhaler technique in the asthma patient population.
The authors have treated the quality of inhaler technique as a nominal variable instead of an ordinal one in the MLR model. Although it has been claimed that MLR could be run for an ordinal variable without violating any assumptions, the appropriateness of using MLR analysis with an ordinal variable may be impacted as the number of independent variables increases or as the categories of the dependent variable increase. Given that there were only a few independent variables and categories, an MLR analysis was conducted.
Overall, the study is well-designed and the authors have identified and acknowledged limitations of the study, which is important for the interpretation of the results. The findings may have important implications for the management of patients with asthma, particularly in terms of improving inhaler technique and hence the efficacy of inhaler therapy.
Reviewer 3 Report
The authors evaluated the prevalence of poor inhaler technique in an asthma patient population and attempted to analyze the impact of various demographic parameters on technique quality. The study concerns an interesting topic with a significant impact on asthma management and has the merit of having used an aerosol inhalation monitor (AIM, Vitalograph®) which is an objective quantitative assessment tool. The study supports that over 2/3 of asthma patients did not use their inhalers appropriately (67% percent had poor or suboptimal technique) across three inhaler devices. The data agree (unfortunately) with previous studies using a different methodology. Some non-significant results may be related to the smallness of the sample analyzed, as the authors themselves acknowledge. There are a few points that could be clarified or better addressed: Line 9, Abstract: I suggest writing "There were SOME significant associations between gender, age, and quality of inhaler technique". Line 129: "The results of the post hoc test showed that four groups were statistically different". Could you specify what are the four groups? Line 137: A bit risky to define a potential association with p value 0.08. Later, in discussion (lines 198-200) the authors write "when all independent variables were included in the multivariate MLR model for the pMDI device, gender was found to significantly affect the quality of the inhaler technique. Men were 2.6 times more likely to have a good inhaler technique with pMDI devices in comparison to women (p < 0.05)". I don't see the multivariate model in the results, it would be better to include it or mention it there. Table 1.1 Participant demographic information and smoking status (n=212). The table is redundant. All presented data are already described in the text (results). Table 1.2. Inhaler device *: What does the asterisk refer to? Table 1.4: 0.019*: The asterisk should explain what the statistically significant difference refers to (poor + suboptimal vs good? poor vs good?). It is unnecessary to point out that p< 0.01 (obvious). Table 1.6: It is not useful to report when p<0.25 (it is not a statistically significant difference: extending the result would correspond to a wrong answer 25% of the time).Author Response
Please see attachment.
